# Explicit Tradeoffs between Adversarial and Natural Distributional Robustness

**Mazda Moayeri**
mmoayeri@umd.edu

**Kiarash Banihashem**
kiarash@umd.edu

**Soheil Feizi**
sfeizi@cs.umd.edu

Department of Computer Science
University of Maryland

## Abstract

Several existing works study either adversarial or natural distributional robustness of deep neural networks separately. In practice, however, models need to enjoy both types of robustness to ensure reliability. In this work, we bridge this gap and show that in fact, *explicit tradeoffs* exist between adversarial and natural distributional robustness. We first consider a simple linear regression setting on Gaussian data with disjoint sets of *core* and *spurious* features. In this setting, through theoretical and empirical analysis, we show that (i) adversarial training with $\ell_1$ and $\ell_2$ norms increases the model reliance on spurious features; (ii) For $\ell_\infty$ adversarial training, spurious reliance only occurs when the scale of the spurious features is larger than that of the core features; (iii) adversarial training can have *an unintended consequence* in reducing distributional robustness, specifically when spurious correlations are changed in the new test domain. Next, we present extensive empirical evidence, using a test suite of twenty adversarially trained models evaluated on five benchmark datasets (ObjectNet, RIVAL10, Salient ImageNet-1M, ImageNet-9, Waterbirds), that adversarially trained classifiers rely on backgrounds more than their standardly trained counterparts, validating our theoretical results. We also show that spurious correlations in training data (when preserved in the test domain) can *improve* adversarial robustness, revealing that previous claims that adversarial vulnerability is rooted in spurious correlations are incomplete.

## 1 Introduction

Despite continuously improving upon state of the art accuracy on various benchmarks, deep image classifiers remain brittle to distribution shifts, suffering massive performance drops when evaluated on non-i.i.d. data. For example, the accuracy of object detectors trained on ImageNet [12] reduces by 40-45% on ObjectNet [5], where images are taken within households at various viewpoints and rotations. The reliance of deep models on *spurious features*, which correlate with class labels but are irrelevant to the true labeling function [30], is one cause of poor model robustness, as performance degrades when spurious correlations are broken. Indeed, model reliance on spurious features like texture [17] and background [69, 6] is well documented. Of greater concern, deep models in safety critical applications such as detection of pneumonia [72] and COVID-19 [11] have been observed to rely on hospital specific spurious markers, causing poor generalization to new hospitals.

Adversarial examples [63, 18] pose another troubling distribution shift, where imperceptible input perturbations can cause model accuracy to drop to zero. Many works have been proposed to improve the adversarial robustness of deep models [36, 74, 52, 59, 37, 10, 66, 55], including the widely popular *adversarial training*, where inputs are augmented via adversarial attack during training [40]. While spurious correlation robustness has also attracted lots of attention [2, 30, 39, 50], the

36th Conference on Neural Information Processing Systems (NeurIPS 2022).

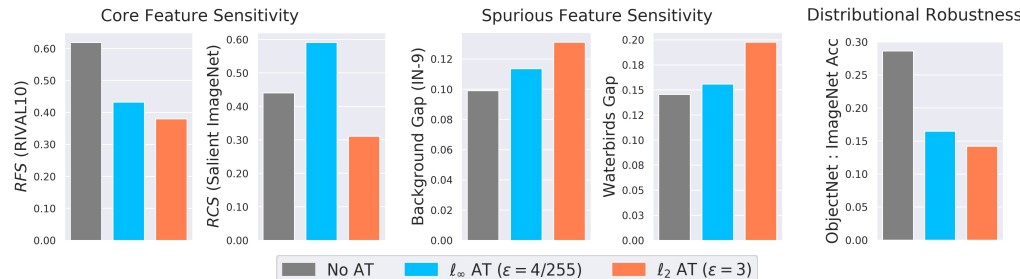

Figure 1: Snapshot of empirical evidence using *RIVAL10, Salient ImageNet-1M, ImageNet-9, Waterbirds,* and *ObjectNet* benchmarks. Results averaged over ResNet18 and ResNet50. **Adversarial training, especially under $\ell_2$ norm, reduces (increases) sensitivity to core (spurious) features. The increased reliance on spurious features leads to worse distributional robustness**.

two problems are most often considered independently, despite both being essential to the safe and reliable deployment of deep models in the wild.

In the few works that do consider adversarial and spurious correlation robustness in tandem, the prevailing argument is that the origin of adversarial vulnerability is that the model focuses on spurious correlations that can be manipulated [77, 73, 26]. However, recently, noise-based analyses on RIVAL10 [42] and Salient ImageNet-1M [61] datasets suggest adversarial training may actually increase model sensitivity to spurious features; a result that is both counter-intuitive and in direct contrast to existing ideology.

To better understand this observation, we first appeal to a simple linear regression problem on Gaussian data with disentangled *core* and *spurious* features. In this setting, we theoretically show

- Adversarial training under $\ell_2$ and $\ell_1$ norms increase model reliance on spurious features, as using spurious features forces an attacker to spread its budget over additional features.
- For $\ell_\infty$ adversarial training, increased spurious feature reliance only occurs when *the scale* of the spurious feature is larger than that of the core features. That is, spurious features are used when perturbations that corrupt core features are too small to disrupt spurious correlations.
- Due to increased spurious feature reliance, there is an *explicit tradeoff* between adversarial and distributional robustness. Specifically, we show that adversarial training decreases model robustness to distribution shifts in the test domain where spurious correlations are broken.

To validate our theory, we evaluate twenty models adversarially trained using $\ell_2$ and $\ell_\infty$ projected gradient descent [51]. Specifically, we inspect performance on multiple spurious robustness benchmarks over synthetic (ImageNet-9 [69], Waterbirds [50]) and real (RIVAL10 [42], Salient ImageNet-1M [61], ObjectNet [5]) datasets. Figure 1 summarizes our experiments, where we find that adversarially trained models consistently show greater sensitivity to spurious features compared to standardly trained baselines, with the effect more dramatic for $\ell_2$ adversarial training than $\ell_\infty$[1]. Finally, we show that the presence of spurious correlations in training data (when preserved in test domain) can improve adversarial robustness, with stronger spurious correlations leading to greater accuracy under attack.

Our work combines two prevalent but often separately considered notions of robustness, yielding surprising theoretically-derived and empirically-supported results. We hope our contributions grant insight to both adversarial and distributional robustness communities, and emphasize the need for holistic evaluations of model robustness.

## 2 Review of Literature

**Adversarial Robustness.** Since adversarial examples were first observed in deep models [63, 18], the phenomenon has been extensively studied. New attacks [8, 35, 43, 15] and defenses [36, 44, 41, 38]

---

[1]High $RCS$ for $\ell_\infty$ AT models is due to reduced *scale* of contextual bias in Salient ImageNet since the data diversity weakens background correlations. In $\ell_1$ and $\ell_2$ adversarial training, models rely on spurious features regardless of their scale. See details in Section 3

are introduced frequently, in a game of cat and mouse where the attacker generally has the upper hand [3]. Certified defenses seek to break this cycle by offering provable robustness guarantees [59, 37, 52, 10]. Arguably the most popular defense is *adversarial training* [40] where images are augmented with adversarial perturbations during training, amounting to a min max optimization.

**Natural Distributional Robustness.** In contrast to synthetic adversarial perturbations, many works seek to characterize the robustness of deep models to naturally occurring distribution shifts, for instance due to common corruptions (noise, blur, etc) [22] or changes in rendition [21, 64]. [31] compiles ten benchmarks of realistic distribution shifts over diverse applications (medical, economic, etc). Many algorithms have been proposed to improve out-of-distribution robustness [39, 50, 2, 34], though in comprehensive evaluations, their gains over empirical risk minimization are marginal, as they often only hold for *certain* distribution shifts [71, 21]. [71] identifies *diversity* and *correlation* shifts as two key dimensions to OOD robustness; our work focuses on the latter.

**Spurious Correlations.** Solely optimizing for accuracy leads deep models to rely on *any* patterns predictive of class in the training domain. This includes *spurious* features, which are irrelevant to the true labeling function. Natural image datasets are riddled with spurious features [32, 60]. Spurious feature reliance becomes problematic under distribution shifts that break their correlation to class labels: sidewalk segmentation struggles in the absence of cars [57], familiar objects cannot be recognized in unfamiliar poses [1] or uncommon settings [28, 5], etc. A natural and ubiquitous spurious correlation in vision is image backgrounds, observed in numerous prior works to be leveraged by models for classification [69, 42] and object detection [48]. Spurious correlations also relate to algorithmic biases [13, 17], with implications for fairness [19, 7, 9, 29], reflecting the importance of this issue.

Accordingly, many works seek to improve spurious correlation robustness. Families of approaches include optimizing for worst group accuracy [50, 24, 49, 75, 39], learning invariant latent spaces [47, 2], appealing to meta-learning [46] or causality [45, 4]. Our work does not focus on mitigation methods, but instead sheds insight on how optimizing for a different notion of robustness (i.e. adversarial) affects spurious feature reliance, and consequently, natural distributional robustness. Generally, models trained under ERM are believed to have a propensity to use spurious features, especially when they are easy to learn, due to bias of learners (algorithmic and human) to absorb simple features first [56] and take shortcuts [16]. However, recent work suggests that core features are still learned under ERM even when spurious features are favored, and simple finetuning on data without the spurious feature can efficiently reduce spurious feature reliance without full model retraining [30].

**Unintended Outcomes of Adversarial Training.** Adversarial training achieves improved accuracy under attack, but comes at the cost of standard accuracy, with multiple works provably demonstrating this tradeoff [74, 14, 23]. Notably, [27] inspires our theoretical analysis, though we focus on the effect of adversarial training on out-of-distribution (OOD) robustness to spurious correlation shifts, rather than its effect on standard accuracy. A more positive outcome is that adversarial training leads to perceptually aligned gradients [54], with applications to model debugging [62, 67], and further, transfer learning on adversarially robust features (despite being less accurate on the original task) yields better accuracy on downstream tasks compared to features learned from standard training [51].

To our knowledge, robustness to adversarial and natural distribution shifts have not been studied in tandem. However, spurious correlations are at times mentioned with adversarial robustness, usually in claims that the origin of adversarial vulnerability is in model's focus on (imperceptible) spurious features [77, 73, 26]. Our results create tension with the contrapositive of their argument, as we show that mitigating adversarial vulnerability (via adversarial training) results in *increased* spurious feature reliance. We do this analytically in a simple linear regression setting (Section 3), and empirically on multiple benchmarks, with an emphasis on natural spurious features (i.e., backgrounds) in our experiments (Section 3). Further, we even demonstrate a case where the presence of a spurious feature leads to *improved* adversarial robustness (Section 4.3)). We note that the spurious features we observe to be positively associated with adversarial robustness may be distinct from those that prior works claim contribute to adversarial vulnerability. However, our result of adversarial training leading to increased spurious feature reliance (of any kind) is novel and contrary to common understanding. Given the critical nature of adversarial and spurious correlation robustness for model security, reliability, and fairness, the significance of our result in revealing potential misconceptions on the interplay of these two crucial modes of robustness should not be understated.

[42] and [61] recently observed decreased core sensitivity on a handful of $\ell_2$ adversarially trained models, in spirit with our findings, but with no explanation. We offer the first rigorous analysis

of this counterintuitive phenomenon, evaluating 5 to 10 times as many models in 5 times as many settings. More importantly, we theoretically prove that adversarial training increases spurious feature reliance, contributing novel fundamental insight as to how optimizing for adversarial robustness can lead to reduced robustness to natural distribution shits, uncovering important effects like the norm of adversarial training and the scale of spurious features at play.

## 3 Theoretical Analysis on Linear models

We begin by analysing the effects of adversarial training on a simple linear regression model. Consider the model $Y = \langle X, \theta^{\text{opt}} \rangle + W$ where $X \in \mathbb{R}^m$ is the input variable, $\theta^{\text{opt}} \in \mathbb{R}^m$ is the optimal parameter, $\langle ., . \rangle$ represents the inner product and $W \in \mathbb{R}$ is a noise variable. We assume that the input variables follow a multivariate Gaussian distribution $N(0, \Sigma)$ where $\Sigma \in \mathbb{R}^{m \times m}$ is the covariance matrix and further assume that $W$ is sampled from the Gaussian distribution $N(0, \sigma_w^2)$. We assume that the set of features $[m]$ consists of two groups, the *core features* $C$ and the *spurious features* $S$. Without loss of generality, we assume that $C = \{1, \ldots, p\}$ and $S = \{p+1, \ldots, m\}$. We assume that the optimal parameter $\theta^{\text{opt}}$ has non-zero entries on the set $C$ only. This implies that the output depends on the input only through the core features and conditioned on the core features, it is independent of the spurious ones. More formally, we assume that $Y \perp X_S | X_C$ where $X_S$ and $X_C$ represent the core and spurious subsets of the input, respectively.

For loss functions, we define the *standard loss* function as $L(\theta) = \mathbb{E}\left[(y - \langle X, \theta \rangle)^2\right]$ ,, and the *adversarial loss* function as

$$L_{p,\epsilon}(\theta) = \mathbb{E}\left[\max_{\|\delta\|_p \leq \epsilon} (Y - \langle X + \delta, \theta \rangle)^2\right], \tag{1}$$

where $p$ is the attack norm and $\epsilon$ represents the norm budget.

We first show an equivalent form of (1) that is more amenable to analysis.

**Theorem 1.** *Assume that $Y = \langle X, \theta^{opt} \rangle + W$ where $W \sim N(0, \sigma_w^2)$ is independent of $X$ and $\theta^{opt} \in \mathbb{R}^m$ is a fixed parameter. Assume further that $X$ follows the distribution $N(0, \Sigma)$ and define $\sigma_\theta^2$ as $(\theta - \theta^{opt})^T \Sigma (\theta - \theta^{opt}) + \sigma_w^2$. The loss function (1) is equivalent to*

$$L_{p,\epsilon}(\theta) = c_2 \cdot \sigma_\theta^2 + (c_1 \sigma_\theta + \epsilon \cdot \|\theta\|_q)^2 \tag{2}$$

*where $c_1 = \sqrt{\frac{2}{\pi}} < 1$, $c_2 = 1 - c_1^2$ and $\|.\|_q$ is the dual norm of $\|.\|_p$, i.e, $\frac{1}{p} + \frac{1}{q} = 1$. Furthermore, the above formulation is convex in $\theta$.*

The above result is similar to Proposition 3.2 in [27] which provides characterization results for the $\ell_2$ norm. The key distinction of our results is providing a simple convex formulation of the robust minimization problem, allowing the results to be easily generalized for an arbitrary $\ell_p$ norm. The theorem shows that the optimal value $\widehat{\theta}$ minimizing the adversarial loss $L_{p,\epsilon}(\theta)$ *is not* $\theta^{\text{opt}}$ and in general, may be non-zero on the set of spurious features $S$. This means that adversarial training directs the model towards using the spurious correlations in order to increase robustness.

The proof of the theorem is provided in the Appendix. The main structure of the proof is similar to that of [27]. We first show that the inner maximization problem of (1) can be rewritten as

$$\max_{\|\delta\| \leq \epsilon} (Y - \langle X + \delta, \theta \rangle)^2 = \left(|Y - \langle X, \theta \rangle| + \epsilon \cdot \|\theta\|_q\right)^2. \tag{3}$$

This allows us to rewrite (1) as

$$\begin{aligned} L_{p,\epsilon} &= \mathbb{E}\left[\left(|Y - \langle X, \theta \rangle| + \epsilon \cdot \|\theta\|_q\right)^2\right] \\ &= \mathbb{E}\left[(Y - \langle X, \theta \rangle)^2\right] + \epsilon^2 \cdot \|\theta\|_q^2 + 2 \cdot \epsilon \cdot \|\theta\|_q \cdot \mathbb{E}\left[|Y - \langle X, \theta \rangle|\right] \\ &\stackrel{(a)}{=} \mathbb{E}\left[\left(\langle X, \theta - \theta^{\text{opt}} \rangle + W\right)^2\right] + \epsilon^2 \cdot \|\theta\|_q^2 + 2 \cdot \epsilon \cdot \|\theta\|_q \cdot \mathbb{E}\left[|\langle X, \theta - \theta^{\text{opt}} \rangle + W|\right], \end{aligned}$$

where for $(a)$ we have used the fact that $Y = \langle X, \theta \rangle + W$. Using the fact that $X$ and $W$ are Gaussian, we show that this is equal to $\sigma_\theta^2 + \epsilon^2 \cdot \|\theta\|_q^2 + 2 \cdot c_1 \cdot \epsilon \cdot \|\theta\|_q \cdot \sigma_\theta$ which we can further simplify to

([2](#)) with some algebraic manipulation. Next, using standard techniques for analysing composition of convex functions, we show that $\sigma_\theta^2$ and $(c_1\sigma_\theta + \epsilon \cdot \|\theta\|_q)^2$ are both convex in $\theta$, implying that $L_{p,\epsilon}(\theta)$ is convex in $\theta$ as well.

Using the convex formulation in Theorem [1](#), we evaluate the linear model for different values of parameters to better understand the reliance of the model on spurious features. In our experiments, we consider a simple model with 5 features, the first two of which are core features.
We let $\eta$ be a parameter controlling the correlation degree between the core and spurious features, with larger values corresponding to higher correlation. For the distribution of the data, define the matrix $\widetilde{Q}$ and $Q$ matrices as

$$\widetilde{Q} = \begin{bmatrix} 1 & \frac{1}{2} & 0 & 0 & 0 \\ \frac{1}{2} & 1 & 0 & 0 & 0 \\ \eta & \eta & 1 & 0 & 0 \\ \eta & \eta & 0 & 1 & 0 \\ \eta & \eta & 0 & 0 & 1 \end{bmatrix}, \quad Q_{i,j} = \frac{\widetilde{Q}_{i,j}}{\sqrt{\sum_{i,j'} \widetilde{Q}_{i,j'}^2}} \tag{4}$$

Note that $Q$ is obtained by normalizing the rows of $\widetilde{Q}$. Each row of the $Q$ matrix corresponds to an input feature. We let $\Sigma$ take the value $QQ^T$. This is equivalent to sampling $X$ from the distribution $Q\mathcal{N}(0, I)$. Throughout our experiments, we set $\sigma_w = 0.1$.

For an arbitrary vector $\theta$, we define its Norm Fraction over Spurious features $\text{NFS}(\theta)$ as

$$\text{NFS}(\theta) = \frac{\sum_{i \in S} \theta_i^2}{\sum_j \theta_j^2} \tag{5}$$

Intuitively, NFS measures the degree to which a model relies on spurious features. For our first experiment, we perform adversarial training with varying parameter $\epsilon$ to obtain a predictor $\widehat{\theta}$ and evaluate its NFS. We consider different choices of the $\ell_p$ norm as well as the correlation parameter $\eta$ The results can be seen in Figure [2](#).

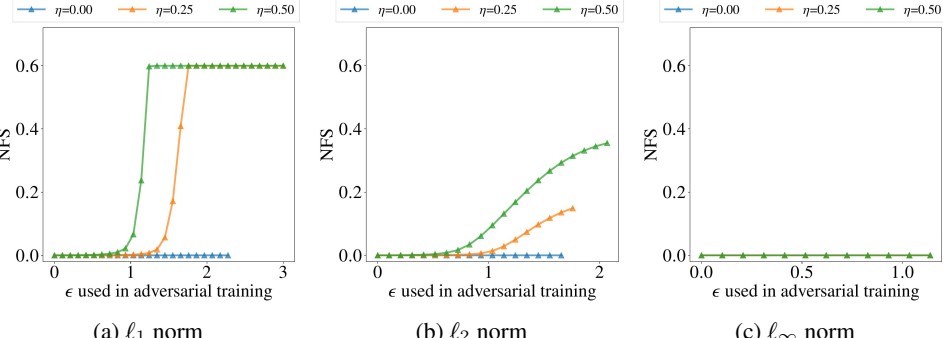

(a) $\ell_1$ norm      (b) $\ell_2$ norm      (c) $\ell_\infty$ norm

Figure 2: Reliance of the adversarially trained model on spurious features as measured by NFS (see Equation ([5](#))) for different choices of $\ell_p$ norm and different values of adversarial budget $\epsilon$ and spurious correlation parameter $\eta$.

As seen in Figure [2](#), for the $\ell_1$ and $\ell_2$ norms, increasing the adversary's budget causes the model to rely more on the spurious features. To understand why this happens, it is helpful to consider a game theoretic perspective: If the model only looks at the core features, then the adversary will only need to perturb these features. Thus, even though the spurious features are normally less suitable for prediction, they now have the advantage of being less perturbed. Assuming $\epsilon$ is large enough, the model will be better off looking at these features in forming its prediction. Of course, if the model only looks at the spurious features, core features will become even more informative as they would be unperturbed as well. In the game's equilibrium, the model would use both the core and spurious features, relying more on the spurious features with increased values of $\epsilon$.

Interestingly, this reasoning does not always apply for the $\ell_\infty$ norm. Indeed, if the model were to only look at the core features, the adversary may still perturb the spurious features with no extra cost. This is because the $\ell_\infty$ norm only measures the *maximum* perturbation in each feature and as long as the perturbations on the spurious features are not larger than the perturbations on the core features,

the norm would not change. The results of Figure 2c support this as the adversary does not rely on the spurious features for its predictions.

Importantly, however, in some cases the adversary may require a larger budget to perturb the spurious features compared to core features. Indeed, if we were to scale a feature by multiplication with a large number, then the adversary would require a larger budget to perturb that feature as the budget for that feature has effectively decreased. We can therefore expect that for the $\ell_\infty$ norm, scaling up the spurious features would cause the model to rely on them for prediction.

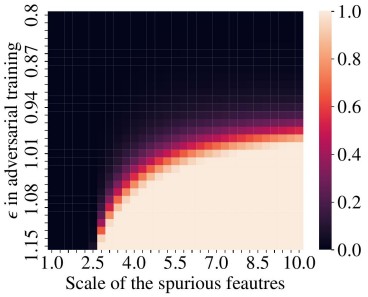

Figure 3 shows that this is indeed the case. The figure shows the norm fraction over spurious (NFS) after scaling the spurious features for different values of $\epsilon$. The scaling is done by multiplying the rows of $Q$ corresponding to the spurious features by a scaling parameter. As seen in Figure 3, larger values of the scaling parameter, as well as larger values of $\epsilon$, cause the model to become more reliant on the spurious features.

Next, we measure the effect of using spurious features on the model's distributional robustness. To do this, we train two adversarial models. The first model, which we denote by "core", uses only the core features while the second model, denoted by "total", uses all of the features. We then simulate a distribution shift that breaks the spurious correlations by adding random Gaussian noise to the spurious rows of matrix $Q$ defined in (4). Specifically, for each entry of $Q$ in a spurious row, we add a noise sampled from $N(0, \sigma_Q^2)$ where $\sigma_Q$ is a noise parameter. We use a fixed value of $\eta = 0.25$ and vary the parameter $\epsilon$, the norm $p$ used in adversarial training as well as the variance of the Gaussian noise. The results are shown in Figure 4.

Figure 3: Norm fraction over spurious features (NFS) with an $\ell_\infty$ constrained adversary as a function of the scale of the spurious features and perturbation budget $\epsilon$.

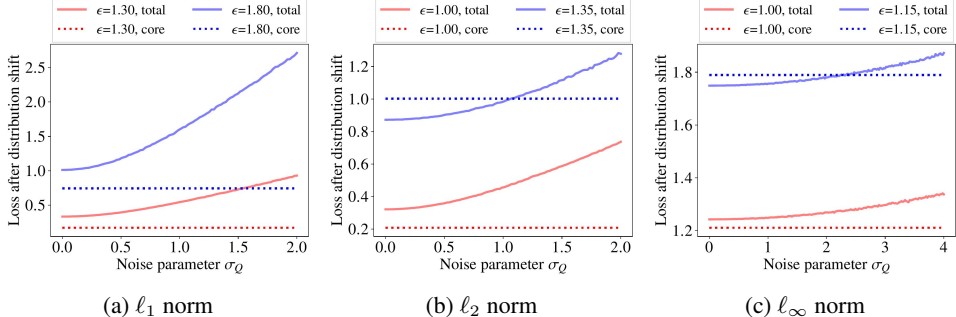

(a) $\ell_1$ norm  (b) $\ell_2$ norm  (c) $\ell_\infty$ norm

Figure 4: Effect of reliance on spurious features on distributional robustness. Each figures compares two models, one using only the core features, and another using all of the features (denoted by "core" and "total" respectively). For the $\ell_1$ and $\ell_2$ norms, the scale of the spurious features is 1 while for the $\ell_\infty$ norm, the scale is set to 3.

We observe that the clean (i.e. in distribution; $\sigma_Q = 0$) loss of the "core" model may be higher or lower than that of the "total" model, but in both cases, the total model is consistently more vulnerable to distributional shifts resulted from breaking spurious correlations.

## 4 Empirical Evidence

We now demonstrate increased spurious feature reliance in adversarially trained models over multiple benchmarks. We evaluate models on two backbones (ResNet18, ResNet50) adversarially trained on ImageNet [12] using two norms ($\ell_2, \ell_\infty$) under five attack budgets (denoted $\epsilon$) per norm, resulting in a $2 \times 2 \times 5 = 20$ model test suite, as well as standardly trained baselines. See appendix for details.

### 4.1 AT hurts Natural Distributional Robustness *only* when Spurious Correlations are broken

We first show reduced distributional robustness of adversarial models occurs specifically in cases where natural spurious correlations are broken. We appeal to the **ImageNet-C** [22] and **ObjectNet**

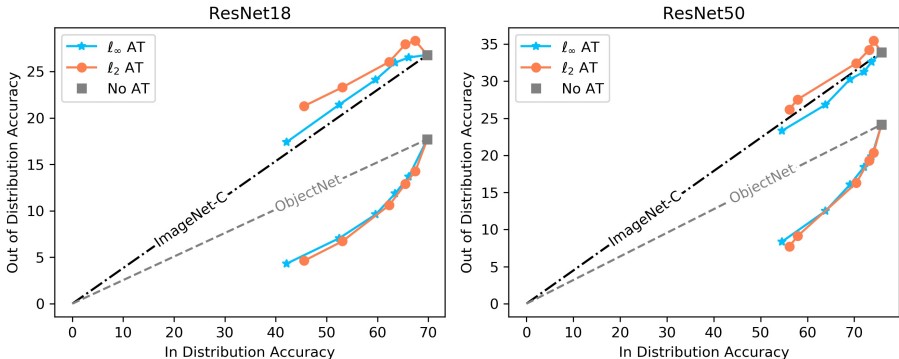

Figure 5: OOD accuracy vs standard ImageNet accuracy for adversarially trained ResNets. ImageNet-C accuracy is closely tied to standard accuracy, but for ObjectNet, where spurious correlations are broken, performance drop is more severe than a linear relation with standard accuracy would entail.

[5] OOD benchmarks. ImageNet-C augments ImageNet samples with common corruptions like noise or blurring, distorting both core and spurious features equally. Crucially, these corruptions do not break spurious correlations. On the other hand, ObjectNet is formed by having workers capture images of common household objects (including samples from 113 classes of ImageNet) *in their homes*. Thus, only spurious features are affected. Namely, ObjectNet introduces distribution shifts in background, rotation, and viewpoint. We plot accuracies on these benchmarks in figure 5.

Recall that adversarially trained models have lower standard accuracy, which can confound our analysis, so we compare the drop in OOD accuracy to the drop in ImageNet accuracy across our model suite. Observe that the ratio of ImageNet-C accuracy to ImageNet accuracy is roughly constant across models. However, the ratio of ObjectNet accuracy to ImageNet accuracy is *lower* for adversarially trained models. Therefore, even after controlling for reduced standard accuracy, the distributional robustness of adversarially trained models is worse than that of standard models. Importantly, this effect does not hold for distribution shifts that maintain spurious correlations, indicating that the reduced distributional robustness is due to increased spurious feature reliance.

## 4.2 Reduced Core Sensitivity, and Difference in the Effect of $\ell_2$ & $\ell_\infty$ Adversarial Training

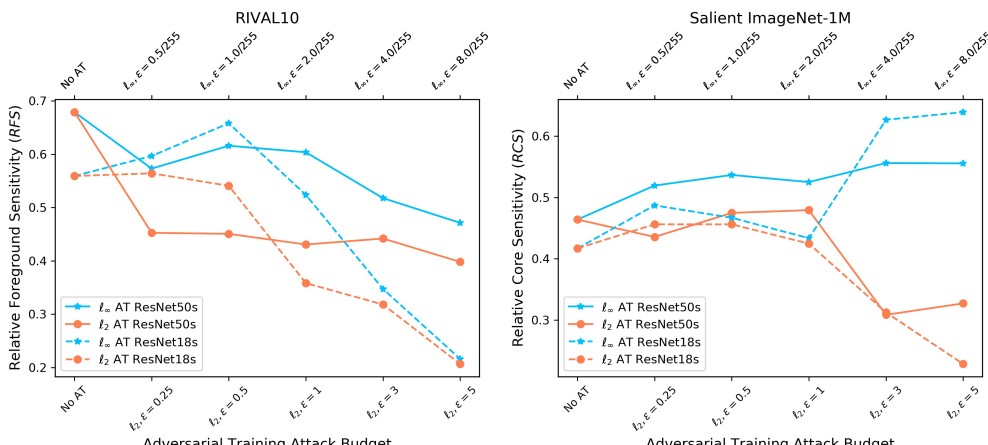

Figure 6: Noise-based evaluation of model sensitivity to foreground ($RFS$ on RIVAL10) or core ($RCS$ on Salient ImageNet-1M) regions. Lower values entail greater sensitivity to spurious regions.

We now directly quantify sensitivity to core features via **RIVAL10** and **Salient ImageNet-1M** datasets [42, 61]. The premise of this analysis is that model sensitivity to an input region can be quantified by the drop in accuracy due to corrupting that region [60]. [42] introduced the noise-based metric *relative foreground sensitivity* ($RFS$), which is the gap between accuracy drops due to background

and foreground noise, normalized so to allow for comparisons across models with varying general noise robustness. RIVAL10 object segmentations allow for $RFS$ computation. Analogously, *relative core sensitivity* ($RCS$) is computed using Salient ImageNet-1M's soft segmentations of core input regions. A key distinction between the two metrics is that $RCS$ is computed directly on pretrained models performing the original 1000-way ImageNet classification task, while $RFS$ first requires models to be finetuned on the **much coarser** 10-way classification task of RIVAL10. Also, Salient ImageNet-1M includes *all ImageNet images*, while RIVAL10 only consists of 20 ImageNet classes.

Figure 6 shows a decrease in $RFS$ and $RCS$ as the attack budget $\epsilon$ seen during $\ell_2$ adversarial training rises. Thus, **adversarial training reduces core feature sensitivity relative to spurious feature sensitivity**. Notably, this effect does *not* hold for models adversarially trained with attacks under the $\ell_\infty$ norm for $RCS$, though it does for $RFS$. Alluding to our theoretical result, we conjecture that in Salient ImageNet-1M, the scales of the spurious features are much smaller than in RIVAL10, due to the diversity of images and finer grain of classes. That is, a smaller perturbation is needed to alter a spurious feature so that it correlates with an incorrect class when there are 1000 classes than when there are only 10 classes with generally disparate backgrounds.

### 4.3 Adversarial Training Increases Background Reliance in Synthetic Datasets

Now, we take a closer look at the reliance of adversarially trained models on the contextual spurious feature of *backgrounds* via the synthetic datasets **ImageNet-9** [69] and **Waterbirds** [50]. Both datasets use segmentations to superimpose objects over varying backgrounds, detailed below.

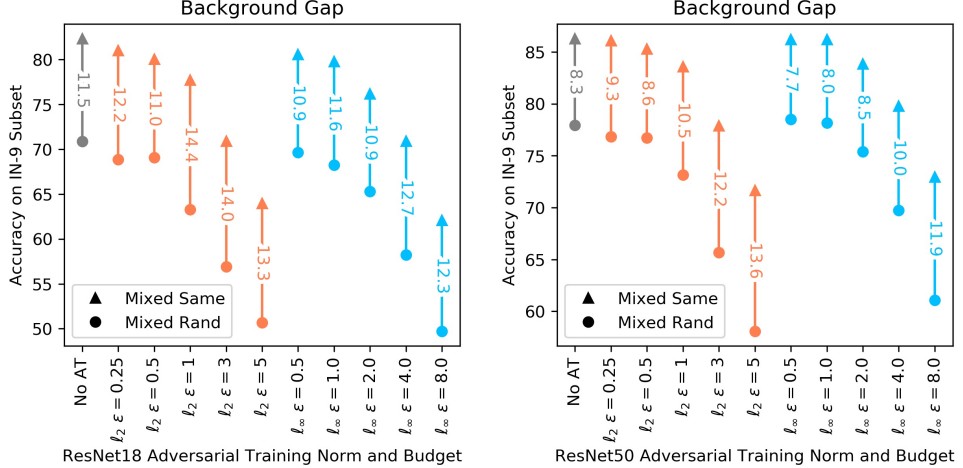

Figure 7: Background Gap (difference in accuracies on ImageNet-9 subsets MIXED-SAME and MIXED-RAND). The drop in accuracy due to background cross-class swapping (MIXED-RAND) causes larger drops in accuracy for robust models, especially $\ell_2$ adversarially trained models.

**ImageNet-9 (IN-9)** organizes a subset of ImageNet into nine superclasses. Multiple validation sets exist for IN-9, where backgrounds or foregrounds are altered; we use MIXED-SAME and MIXED-RAND. In both sets, original backgrounds are swapped out for new ones. Crucially, in MIXED-SAME, the new backgrounds are taken from other instances *within the same class*, while MIXED-RAND uses *random backgrounds*. The metric, **Background Gap**, is the difference in model accuracy on MIXED-SAME and MIXED-RAND (i.e. drop due to breaking spurious background correlation).

Figure 7 shows that the background gap for our test suite of twenty robust ResNets and two standardly trained baselines. Nine out of the ten $\ell_2$ adversarially trained models have larger background gaps than the standard baselines, while the same is true for five out of the ten $\ell_\infty$ adversarially trained models. When considering relative gaps (i.e. as a percent of the accuracy on MIXED-SAME), the increase in gap becomes even more dramatic, with the $\ell_2$ adversarially trained ResNet50 for $\epsilon = 5$ having a $18.9\%$ relative drop, compared to the $9.7\%$ relative drop for the standardly trained ResNet50.

**Waterbirds** combines foregrounds from Caltech Birds [65] and backgrounds from SUN Places [68]. The task is binary classification of land birds and water birds. The majority group ($95\%$ of training

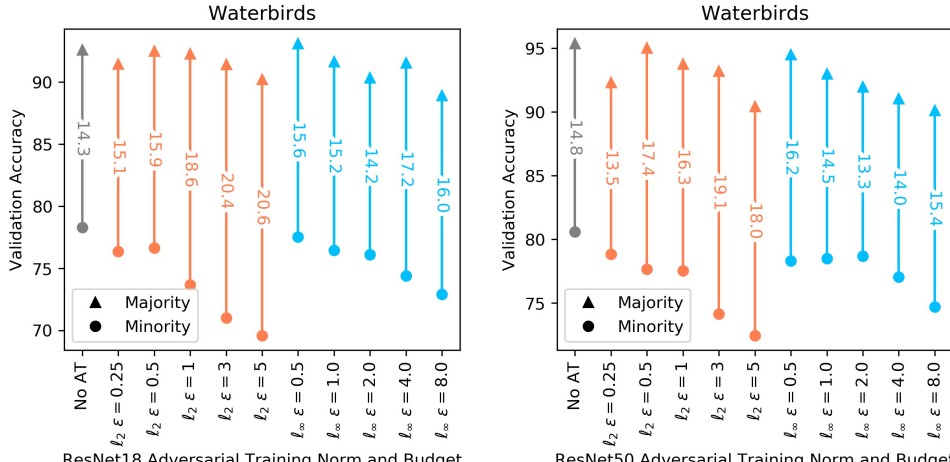

Figure 8: Accuracies on Waterbirds subsets where spurious correlation is intact (majority group; e.g. land birds on land backgrounds) and where it is broken (minority group; e.g. land birds on water backgrounds). The drop in performance due to breaking the background correlation grows larger for robust models, especially $\ell_2$ adversarially trained models.

samples) consists of land birds over land backgrounds and water birds over water backgrounds. The minority group breaks this spurious correlation, placing land birds over water backgrounds, and vice versa. The test set is evenly split between these groups. We train only a final linear layer atop the frozen feature extractors (so that models remain adversarially robust) for each of our models on the Waterbirds training set for ten epochs, saving the model with highest validation accuracy.

Figure 8 shows majority and minority group accuracies, and the gap between them, for our test suite of models. Again, we see increased gaps for robust models, with $100\%$ of $\ell_2$ and $60\%$ of $\ell_\infty$ models respectively having larger gaps than the standardly trained baseline on the corresponding backbone.

In both benchmarks, breaking the background spurious correlation causes a more significant drop in performance for adversarially trained models than standardly trained models, indicating that adversarial training led to increased reliance on backgrounds. The observed affect is stronger for $\ell_2$ adversarially trained models than $\ell_\infty$ ones. Further, the gaps grow near monotonically with $\epsilon$.

### 4.4 Reverse Effect: Presence of Spurious Correlations Can *Improve* Adversarial Robustness

Finally, we show evidence that is directly at odds with the claim that spurious features lead to adversarial vulnerability. We train ResNet18s on CIFAR10 [33] with a spurious feature injected. Namely, images have all values in one color channel slightly increased. The majority group consists of red-shifted images from classes $0-4$ and green shifted images from classes $5-9$, while the minority group has reverse color-shifts. The parameter $\rho$ is the ratio between majority and minority group size, controlling the strength of the spurious correlation (higher $\rho$ means stronger spurious correlation; $\rho = 1:1$ means the spurious feature has no predictive power). We then evaluate the accuracy of the trained model under adversarial attack on an i.i.d. test set (**spurious feature retained**, no distribution shift).

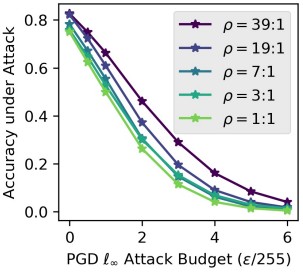

Figure 9: Injecting spurious feature improves adversarial robustness.

Figure 9 visualizes the results. Not surprisingly, the clean accuracy is higher for models trained on data with higher $\rho$, as the spurious feature is more predictive for higher $\rho$. In fact, the added predictive influence the spurious feature leads to better accuracy *under attack*, with the gap between highest and lowest $\rho$ values growing up to four fold compared to the baseline gap in clean accuracy. Thus, using a spurious feature can improve adversarial robustness. Despite being on a contrived example, this experiment shows that, while some spurious correlations may cause adversarial vulnerability, *others do the opposite*: the picture is more nuanced than previously assumed.

## 5   Acknowledgements

This project was supported in part by NSF CAREER AWARD 1942230, HR001119S0026 (GARD), ONR YIP award N00014-22-1-2271, Army Grant No. W911NF2120076, the NSF award CCF2212458, an AWS Machine Learning Research Award.

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
