# OpenReview forum: "Explicit Tradeoffs between Adversarial and Natural Distributional Robustness"
_NeurIPS.cc/2022/Conference — NeurIPS 2022 Accept_

### Official Review · Reviewer_gNwp · 2022-06-16

**Rating:** 5
**Confidence:** 4
**Soundness:** 3 good
**Presentation:** 3 good
**Contribution:** 2 fair

**Summary:**

The authors investigate the connection between adversarial robustness and reliance on spurious features.  They demonstrate that adversarial training with respect to Lp-based threat models increases reliance on spurious features.  They provide theoretical analysis of a simple linear regression setting and provide experimental results on image datasets to demonstrate that adversarially robust models have higher reliance on spurious features, leading to lower natural distributional robustness.

**Questions:**

- One point of confusion I had was that the authors state that Theorem 1 implies that adversarial training directs the model towards using spurious features (lines 154-156).  How exactly Theorem 1 implies this is unclear to me, could the authors clarify this point?
- In equation 5, what is P?
- In the toy experiments in section 3, wouldn't the dependence on core vs spurious features for an adversarially robust model depend on the number of features that a considered core vs spurious?  In the toy example, there are 2 core features vs 3 spurious features.  I think that in the setting where there are more core compared to spurious features, then we would not see the jump in spurious feature reliance as we do in the toy examples.
- Following the proof, shouldn't equation (3) be $\text{max}_{||\delta||\le \epsilon} (Y - \langle X + \delta, \theta \rangle)^2 = (|Y - \langle X, \theta \rangle| + \epsilon ||\theta||_q)^2$?
- In the proof of Theorem 1, in proving equation 3, it states that "with a suitable choice of $\theta$" the equality (b) holds, but the statement in equation 3 should hold for any $\theta$ when it is used when rewriting equation 1, since loss is evaluated for any given $\theta$, which may not be the specific value of $\theta$ such that exact equality holds.  Should this be "with a suitable choice of $\delta$ instead?  I think this also goes for the equality of the triangle inequality portion too, since we are choosing the max over $\delta$, we would choose $\delta$ to achieve the upper bound rather than putting restrictions on $\theta$.
- Also in the sketch proof of Theorem 1 in main text, in the derivation after line 159, I believe that you are missing a $||\theta||_q$: ie) in the 2nd line the last term should be $2 \epsilon ||\theta||_q \mathbb{E}[|Y - \langle X, \theta \rangle|]$, but the $||\theta||_q$ is missing.  This error is also present in the proof in the appendix and carries down for many steps of the proof until the very last step in the proof of equation (2).
- in after 576 of the Appendix, in the equality marked by (a), the last term is missing $c_1$.  This also carries down to the next step in the proof of equation (2).

**Strengths And Weaknesses:**

Strengths:
- connects 2 directions of research (natural distributional robustness and adversarial robustness) which to the best of my knowledge has not been studied in depth in prior work
- experiments are in depth

Weaknesses:
- While the authors claim that adversarial robustness increases reliance on spurious features and frame this as a surprising result, I'm actually not surprised by this.  I think it really depends on what spurious features are being considered.  Adversarial training increases robustness against certain spurious features specifically small high frequency changes (ie. single pixel change), but for other existing spurious features which are considered in this paper, I do not see why adversarial training would suddenly prioritize using core features over the many other spurious features (ie background and color correlations) which may be easier to learn.  Because of this, I'm not very convinced about the significance of this work.
- Given that generally we are interested in classification problems in adversarial ML, I would have been more excited to see theoretical results for a binary classification problem rather than regression problem, but this is a minor point
- mathematical typos in the proof sketch in the main text and proof in the appendix make proof hard to follow (see questions).

---

> ### Author Response · Authors · 2022-08-02
> **Author response**
>
> We thank the reviewer for their comments and taking the time to closely read our work. We address the concerns below.
>
> **"While the authors claim that adversarial robustness increases reliance on spurious features and frame this as a surprising result, I'm actually not surprised by this. … Adversarial training increases robustness against certain spurious features … I do not see why adversarial training would suddenly prioritize using core features over the many other spurious features (ie background and color correlations) which may be easier to learn. "**
>
>  While we wholeheartedly agree that the exact kind of spurious feature considered is of utmost importance, we respectfully disagree that our result is not surprising. First, we clarify that we do not say that prior works claim adversarially trained models are expected to ‘prioritize learning core features over spurious ones’. We instead state that most prior work does not engage spurious correlation robustness and adversarial robustness together. Further, when those two notions of robustness are mentioned together, it is usually in the sense that the model reliance on (certain) spurious features is the cause of adversarial vulnerability (see lines 108-122). Our result directly contradicts the contrapositive of the aforementioned common claim, which is why we find it surprising. We kindly note that all three other reviewers also find this interplay surprising (RU2bQ: “significant and thought provoking”, R1sj: “bizarre”, Rmc4G: “counterintuitive”).
>
> As mentioned, we completely agree with the reviewer’s argument that the type of spurious feature matters, and hope that our result encourages the community to not treat spurious correlation robustness (or robustness entirely) as a problem with a one-size-fits-all solution. Without paying attention to the nuance involved with a specific distribution shift at hand, significant misconceptions may arise, such as the one we extensively cover (i.e. incorrectly assuming that adversarially trained models are generally more robust than, or at least as robust as, standard ones).
>
> **Typos:** We thank the reviewer for their careful read of our work, and have corrected all mentioned typos in the updated version. We note however that both the result and its proof (barring the mentioned typos) are correct.
>
> **Number of core vs spurious features:** The number of core vs. spurious features does indeed have an effect on the degree to which adversarially trained models rely on spurious features. However, it does not change the fact that adversarially trained models rely on spurious features to any (non-zero) degree, while standardly trained models do not (in our theoretical setting). In fact, in appendix C, we explicitly characterize and also show in simulation that the amount of weight placed on spurious features for $\ell_1$ adversarially trained models is precisely equal to the ratio of the number of spurious features to the number of total features (Figure 10a). We have repeated these experiments for the $\ell_2$ norm and obtained results that are qualitatively similar to those of Figure 2 (see Figure 11 in the revised draft). While the exact degree of reliance on spurious features varies with the number of spurious features, in all cases, the adversarially trained models make non-zero use of the spurious features.
>
> **"How exactly Theorem 1 implies this is unclear to me"**
>
>  Theorem 1 shows that the optimal parameter $\theta$ for the adversarial objective will have non-zero weights along spurious features, as the second term in the loss function (eq 2) involves the q-norm of $\theta$. Thus, optimizing for the adversarial objective forces a balance between having $\theta$ be as close to $\theta^\text{opt}$ as possible and $\theta$ having as low q-norm as possible. It may be necessary to diverge from the $\theta^\text{opt}$ (i.e. shift weight from core features to spurious ones) so that the total objective is minimized. This is further validated by the results in Figures 2 and 3 which use the characterization in theorem 1 to demonstrate that the adversarially trained model relies on the spurious features (non-zero NFS values). Note that a standardly trained model, under our theoretical setting, will exactly recover $\theta^\text{opt}$; that is, they will make no use of spurious features. Thus, any new use of the spurious features is due to adversarial training.
>
> If the reviewer believes our comments and revisions address their concern, particularly around the significance of our result, we would greatly appreciate if they can consider raising their score, as our result may be more surprising to more readers than the reviewer may expect. Thank you!

---

> > ### Comment · Reviewer_gNwp · 2022-08-04
> > **Thank you for the clarifications**
> >
> > Thank you for the clarifications.  I think most of my questions are addressed, but I'm still not fully convinced about the first point.  It think my comment is similar to that of Reviewer R1sj which is that adversarial vulnerability could still stem from spurious features and adversarial training increases robustness against those *specific* spurious features, and not necessarily other ones.  However, there is not much discussion of this in the paper.  I also do not think that the authors results really contradict the claims made by prior works: "the origin of adversarial vulnerability is in model’s focus on (imperceptible) spurious features" (lines 109-110) due to the word "imperceptible" here.  In the experiments, the authors look at specific spurious features (ie. background and color), which are not necessarily the imperceptible spurious features that are suggested to lead to adversarial vulnerability (which is also why I didn't find the results very surprising).  I would appreciate if the authors revised some of the wording in the paper (specifically lines 111-112 and 294-295) to remove the statement that their results contradict previous findings.
> >
> > I appreciate the clarification of theoretical results and additional experiments on varying number of spurious features and will update my score to 5.

---

> > > ### Author Response · Authors · 2022-08-08
> > > **Thank you for valuable discussion, Additional Revisions Made; More To Come if Accepted**
> > >
> > > Thank you very much for taking the time to read our rebuttal and increasing your score. To the point regarding the significance of our observation, we have made additional revisions to the manuscript in the lines you specified, so to remove claims that we explicitly contradict prior work. We note that the prevalent arguments linking spurious correlations to adversarial vulnerability may lead readers to infer that these arguments hold without paying close attention to the type of spurious correlation at play. We hope our paper will encourage the community to be more mindful of the importance of the specific type of distribution shifts a proposed method aims at improving, as well as considering the effect of their method on other types of distribution shift.
> > >
> > > Moreover, our theoretical setting reasons about spurious feature reliance from a purely mathematical sense, wherein spurious features can be soundly defined (i.e. as conditionally independent of core features with respect to the class label). By considering a simplified setting where core and spurious features are disentangled, we are able to preclude discussion of the specific type of spurious correlation involved. In this setting, we show adversarial training increases spurious feature reliance (for a sufficiently large attack budget and specific norms, such as l1 and l2). This theoretical result disputes the contrapositive of the prevalent argument that spurious feature reliance reduces adversarial robustness, as adversarial training increases adversarial robustness, but in turn leads to heightened spurious feature reliance. We highlight that this theoretical result makes no assumption about the type of spurious correlation at play. While this is a consequence of the assumptions in our setting, we believe the result is surprising and insightful, particularly when paired with extensive experiments.
> > >
> > > If our paper is accepted, we will make use of the extra content page to have a discussion on the nuances tackling the many faces of robustness, which perhaps are often overlooked/oversimplified. We thank you for your valuable comments and will certainly directly incorporate your feedback in the extra page if given the opportunity.

---

### Official Review · Reviewer_mc4G · 2022-07-11

**Rating:** 6
**Confidence:** 3
**Soundness:** 3 good
**Presentation:** 3 good
**Contribution:** 3 good

**Summary:**

The authors claim that the adversarial training encourages to model to reply more on spurious features, which hurts the robustness under distributional shift. To support their claim, they proposed a simplified setting: a linear model with some standard core and spurious features. They found the adversarial training increased the model weights on spurious features. Finally, a few synthetic/real datasets are studied to verify the hypothesis empirically.


**Questions:**

First, although the dichotomy of “core” features and “spurious” features are mentioned many times in previous works, it would be great to have some formal definitions in theory and some examples. In image recognition, is it true that “foreground” means “core features” and “background” means “spurious features”?

Secondly, I find the definition of “distribution shift” in this paper a bit restrictive in that it only changes the “spurious features” (L210 - L213). But in reality, both feature sets will be affected.

Thirdly, the results under the Linf constraint seem to be inconsistent: sometimes it behaves just like L2 constraint, otherwise, it behaves as the theory implies. For example, in Figure 5 the OOD accuracy drops as quickly as L2 constraint; while Figure 6 shows the RCS doesn’t change.

Finally, if we think the other way round: suppose we train the classifier with a distributionally robust optimizer. Then according to this paper, the model should only use the core features, thus even more susceptible to adversarial examples than the original setting. But I don’t think it will be observed in real experiments.


**Limitations:**

Please refer to the section above.

**Strengths And Weaknesses:**

Overall, the paper made a counter-intuitive claim that adversarial robustness hurts distributional robustness; and this is due to the shifts in weights from the core feature set to the spurious feature set. The theoretical analysis, although seems a bit over-simplified, correlates well with almost all the empirical results. The experimentations are well designed and convincing, and I have no difficulty to follow.

---

> ### Author Response · Authors · 2022-08-02
> **Author response**
>
> We thank the reviewer for their insightful comments and suggestions. We address the questions below.
>
> **Fuzzy definition of ‘core’ and ‘spurious’:** Mathematically, we define spurious features as ones that correlate with the class label, but when conditioned on core features, are independent of the class label (line 141-142). In practice, for image classification, as mentioned by the reviewer, we do assume that any pixels containing the object (i.e. the foreground as used in RFS) are core features, and other pixels are spurious features. We will further emphasize this point in our paper.
>
> **Restrictive definition of distribution shift:** First, we note that while some distribution shifts affect both core and spurious feature sets, there are many interesting distribution shifts where only spurious features are affected. For example, when objects appear in unusual environments, the object itself (and hence all core features) remains unchanged, while only the background (spurious feature) is out of distribution. We agree that the term ‘distributional robustness’ is very broad, and while we explicitly specify that the notion of natural distributional robustness related to our work is spurious correlation robustness (Lines 12, 52, 103, 220, 228), we will make further modifications to clarify this point.
>
> **Inconsistent results for $L_{\infty}$ attack:** Our empirical results show that $\ell_\infty$ adversarial training only sometimes leads to increased spurious feature reliance, unlike $\ell_2$ adversarial training, which consistently increases spurious feature reliance. Note, however, that this falls in line with our theoretical predictions, as we observe a special dependence on the scale of spurious features for $\ell_\infty$ adversarial training that is not present for $\ell_2$ or $\ell_1$ training. As shown in Figure 3, with a fixed attack budget (e.g. eps=1.1), some values for the scale of the spurious feature can lead to total reliance on spurious features (e.g. scale=5), while others lead to no reliance (e.g. scale=1). We refer the reviewer to lines 188-205 for more explanation.
>
> **"suppose we train the classifier with a distributionally robust optimizer. Then according to this paper, the model should only use the core features, thus even more susceptible to adversarial examples than the original setting. But I don’t think it will be observed in real experiments."**
>
>  While our central claim that adversarial training can lead to increased model reliance does not necessarily require the suggested claim (i.e. that distributionally robust optimizers reduce adversarial robustness), we nonetheless explore an somewhat related instance of this suggested claim in Section 4.4: we show that a model that does not rely on spurious features (because there is no spurious correlation when the ratio of majority correlation-preserving group to minority correlation-breaking group is $\rho=1:1$) indeed has lower accuracy under adversarial attack than the model that does use the spurious feature. We consider the reviewer’s suggestion to be a great direction for future work, emphasizing the spirit that the robustness community should strive to work holistically; that is, regardless of the specific notion of robustness one seeks to improve, they should additionally evaluate on other notions of robustness. By identifying an unexpected interplay between adversarial and spurious correlation robustness, we hope our work encourages the field to embrace this notion of holistic robustness evaluations.
>
> If the reviewer feels that we have clarified any concerns, we would greatly appreciate any raise in score.

---

> > ### Author Response · Authors · 2022-08-08
> > **Hope we have clarified your concerns**
> >
> > Hello, we thank the reviewer again for all of their feedback. We just wanted to gently check in to see if our rebuttal sufficiently addressed the reviewer's questions. If so, any raise in score would be extremely appreciated, as we may be somewhat on the borderline of acceptance at the moment.
> >
> > If there are any lingering questions, we'd be more than happy to answer. Thank you!

---

### Official Review · Reviewer_U2bQ · 2022-07-11

**Rating:** 7
**Confidence:** 4
**Soundness:** 4 excellent
**Presentation:** 4 excellent
**Contribution:** 4 excellent

**Summary:**

This paper offers an analysis of the explicit tradeoffs between adversarial robustness and natural distributional robustness. The analysis well explains the phenomenon that adversarial training sometimes increases model sensitivity to spurious features in experiments. The reason is that adversarial training may actually increase model reliance on spurious features.

**Questions:**

- Adversarial training has been extended to improve model robustness beyond $L_p$ balls, such as [1][2]. It is then natural to wonder whether a method like [1] can increase model reliance on spurious features and thus decrease distributional robustness. After all, the methods like [1][2] are somewhat designed to improve distribution robustness.

[1] Laidlaw, Cassidy, Sahil Singla, and Soheil Feizi. "Perceptual Adversarial Robustness: Defense Against Unseen Threat Models." International Conference on Learning Representations. 2020.
[2] Lin, Wei-An, et al. "Dual manifold adversarial robustness: Defense against lp and non-lp adversarial attacks." Advances in Neural Information Processing Systems 33 (2020): 3487-3498.

**Limitations:**

See above.

**Strengths And Weaknesses:**

**Pros:**
- This paper is the first to theoretically show that adversarial training may increase model sensitivity to spurious features, though some researchers implicitly supposed that adversarial training would always decrease model reliance on spurious features.
- A corollary of the analysis is that adversarial training may harm model robustness to the distribution shifts that break the spurious correlations. This result is significant and thought-provoking.

**Cons:**
- Section 3 only gives one analytical result showing a simplified version of the adversarial loss. This result is good, which indeed implies that the optimal $\theta$ may be non-zero on the set of spurious features. However, the theoretical analysis fails to show how much adversarial training can increase model reliance on spurious features. The paper then relies on verbal reasoning to reach the remaining theoretical results, without further formulating clearer theorems or corollaries. Thus, the so-called "explicit" tradeoffs between adversarial and distributional robustness are not so obvious.

---

> ### Author Response · Authors · 2022-08-02
> **Author response**
>
> We thank the reviewer for their comments on the novelty and significance of our work, as well as offering an intriguing follow up analysis. We discuss specific comments below.
>
> **On Exactly Quantifying Spurious Feature Reliance and Loss in Distributional Robustness:** The degree to which adversarial training increases the reliance on spurious features varies due to many underlying factors, such as the type of adversarial training (i.e. threat model, attack budget during training) and the data distribution (i.e. correlation between spurious feature and class label, type and number of spurious features, etc). With our theoretical analysis, we intended to explicitly show a situation in which adversarial training would lead to usage of spurious features, while standard training would not. To highlight this result, we do not formulaically engage the numerous factors that determine the exact amount that spurious feature reliance increases due to adversarial training. Instead, we provide some discussion of these factors, and further, include simulations in a setting where we can directly quantify the amount that spurious features are relied upon (see Figure 2 and Figure 10 in appendix). Furthermore, we show that when spurious features are corrupted with noise, adversarially trained models that use spurious features become less and less accurate (Figure 4), demonstrating the reduced robustness to distribution shifts that break spurious correlations. We opted not to more precisely characterize the exact tradeoffs with theorems, as it would require the knowledge of some quantities (i.e. number of core/spurious feature, scale of spurious feature, etc) that are not currently accessible in practice. However, we hope our work shows the importance of these quantities, and perhaps encourages others to develop tools to identify and measure the presence of core and spurious features in their data, and the correlations between them.
>
> **On non-$\ell_p$ adversarial training:** We thank the reviewer for this question, and find it extremely interesting. In this work, we have chosen to focus on $\ell_p$ robustness as it is the most commonly considered notion of robustness. We agree however that exploring different notions of robustness is an excellent direction for future work as it may lead to interesting and counterintuitive results. Indeed, as shown in our paper, even the distinction between different $\ell_p$ norms (e.g., $\ell_2$ vs $\ell_\infty$) can have significant effect on spurious feature reliance, so we find it very plausible that a carefully designed adversarial training procedure can perhaps achieve both adversarial and distributional robustness.

---

> > ### Comment · Reviewer_U2bQ · 2022-08-09
> > **Thanks**
> >
> > I appreciate the authors' response. Though this work does not characterize the tradeoffs with precise theorems, it indeed shows explicit tradeoffs via thorough discussions. Moreover, this highlights the importance of the tradeoffs, which would be of interest to the community. Thus, I would like to retain my score and recommend acceptance.

---

### Official Review · Reviewer_R1sj · 2022-07-11

**Rating:** 5
**Confidence:** 4
**Soundness:** 3 good
**Presentation:** 2 fair
**Contribution:** 3 good

**Summary:**

This paper argues that adversarial robustness might be at odds with robustness with natural distribution shifts. To that end, the authors first illustrate this intuition on a synthetic setting consisting of a linear regression model trained on Gaussian data with controlled spurious correlations. In this setup, they show numerically that the adversarially trained models tend to rely more on the spurious correlations than their standard couunterparts, and have worse robustness agains certain distributions shifts. Afterwards, the paper presents several evaluations of an adversarially trained model suite on multiple distribution shift benchmarks, including real-world ones and synthetic ones. The experiments on these model seem to agree with the numerical simulations presented in the prior section, i.e, adverarially trained models rely more on certain spurious correlations.

**Questions:**

I have my concerns accepting the arguments in Sec. 4.4. In particular, the authors claim "[...] this experiment shows that exclusively linking spurious correlations to adversarial vulnerability is a fallacy". However, I do not think this experiment shows that. At most, it vaguely discards that certain color spurious correlations are the root cause of adversarial vulnerabilities, but there could exist thousand other types of spurious correlations that could explain the existence of adversarial examples. Bearing this in mind, I would appreciate a further discussion from the authors on this particular point in which they argued why their work demonstrates that adversarial examples cannot be due to any form of spurious correlation.

I would also appreciate if the authors could further comment on the novel aspects of their work on top of the observations in [38] and [55].

**Limitations:**

I see no clear negative societal impact stemming directly from this work. Furthemore, the authors provide a short societal impact statement in the appendix.

**Strengths And Weaknesses:**

## Strengths

1. **Strong and clear message**: I really appreciate the core message of the paper which tries to show there is an odd and counterintuitive relation between $\ell_p$ adversarial robustness and reliance on spurious correlations. This is a valuable insight for the community, as it reminds us of the bizarre dynamics that $\ell_p$ adversarial training has on the feature extraction power of neural networks.
2. **Good balance between synthetic experiments and practical evaluations**: Overall, I find the numerical simulations on the synthetic data presented on Fig. 2 and Fig. 3 quite interesting, and they nicely complement the main observations provided on ObjectNet, RIVAL10, ImageNet-9, and Waterbirds.

## Weaknesses

1. **Novelty**: My main concern is with regards to the added value that these observations provide to the community given what is already known from prior literature. In particular, as described in the introduction, both [38] and [55] had already observed a stronger reliance on background spurious features for adversarially trained models, which means that the main new insight from this work is the observation that the same increase in spurious correlation also happens in ObjectNet, ImageNet-9 and Waterbirds.
2. **Some hand-wavy arguments**: In relation to the previous point, I understand that the main objective of this work is therefore to provide some insights on why adversarially trained models are less robust to distribution shifts. In this regard, although I find the numerical simulations in Sec. 3 quite interesting, I do not think they can answer the question of why the same effect happens on real data. All in all, the spurious correlations this work alludes to are mostly semantic, e.g., backgrounds, or object positionings, and they are hard to map to the geometric concepts in Sec. 3. This makes some of the arguments regarding the "strength of different spurious features", or the reasons why we do not see the same effects on ImageNet-C or for $\ell_\infty$ models, quite hand-wavy as they are not well-defined, nor rooted on any quantifiable evidence.
3. **Clarity (minor)**: Some passages in this manuscript feel a bit rushed and are hard to read, e.g., the presentation of the experiments leading to Fig. 4. The writing, in generall, could be more polished, and the paper be better formatted. In this sense, I believe that making the presentation of Fig. 5, Fig. 7 and Fig. 8 consistent with be nice.

---

> ### Author Response · Authors · 2022-08-02
> **Author response**
>
> We thank R1sj for their insightful comments.
>
> 1. **On Novelty**: We make several novel and significant contributions over [38,55]. First, we formulate a theoretical setting in which we can carefully study how factors like the norm of adversarial training, attack budget, and data conditions can affect the degree to which adversarial training increases model reliance on spurious features. In this setting, unlike prior work, we provably show that adversarial training can increase reliance on spurious features, characterize this increased reliance numerically, and gain novel insights. For example, we discover that $\ell_\infty$ adversarial training does not suffer from increased spurious feature reliance as consistently and severely as $\ell_2$ and $\ell_1$ adversarial training. We also find that increased spurious feature reliance only occurs for sufficiently large attack budgets during training (Figure 2). Second, our empirical study is far more extensive than prior work. Those works take a single approach to measuring spurious feature sensitivity (via the noise based metrics RFS and RCS) on a handful of $\ell_2$ adversarially trained models. In fact, those metrics have not been validated by other more typical measures of spurious feature reliance. In contrast, our work studies a total of 35 adversarially trained models over 8 architectures (we repeat all experiments in the main text on 15 additional models over six new architectures in the appendix), while engaging four new benchmarks that are much more standard than RCS and RFS. In summary, our work provides the first rigorous and focused analysis of the counterintuitive interplay between adversarial robustness and spurious correlation robustness.
> \
> The prior works, on the other hand, primarily focus on contributing datasets and benchmarks for measuring spurious feature reliance, making passing observations regarding many types of models (of which adversarially trained models make up only a small subset). While observations about adversarially trained models in those works inspire our study, we go far beyond the initial observation, answering questions like *why does this occur* (see lines 179-187), *when does this occur*, with theoretical analysis that allows for characterization of the effect of many relevant factors, and a significantly more extensive set of experiments using more standard spurious correlation benchmarks.
>
> 2. **Strength of Arguments**: As this is the first work to formally consider the trade-off between adversarial and distributional robustness, we have chosen a simplified theoretical model to obtain an analytical solution. We note that our setting is similar to [Khani et al. ], who also study a linear model atop disentangled core and spurious features to provide theoretical insight.
> \
> While we agree with the reviewer that this setting is simpler than the one considered in Section 4, our work shows a roadmap towards more complex analysis (e.g. by extending to more complex data distributions) which we leave for future work. We address more specific critiques in the next comment.
>
> 3. **Clarity**: We thank the reviewer for their close read of our work, and have revised the text for clearer reading, including making the color scheme of Figures 5, 7, 8 consistent, as requested.
>
> 4. **Section 4.4**: The intention of the experiment in Section 4.4 is not to show that ‘adversarial examples cannot be due to any form of spurious correlation’, but instead to refute (via a counterexample) the claim that spurious feature usage necessarily leads to adversarial vulnerability. We completely agree that the exact type of spurious correlation is of utmost importance with respect to its effect on adversarial robustness (see line 116-117), though we find that most common works overlook this nuance. A central goal of our work is to shed insight into the complex and sometimes contradictory nature of various notions of robustness. Thus, we believe claims that exclusively link adversarial vulnerability to the cause of spurious feature reliance (a prevalent idea within the adversarial robustness community) are not completely true because they overlook the precise nuance you mention: that the type of spurious correlation matters. Indeed, spurious feature usage is often scapegoated as the cause for poor generalization to various domain shifts, though we show that at least in one case, using spurious features as context can indeed assist in generalization (i.e. to adversarial distribution shifts)! We have revised the language of this section to clarify our intended message.
>
> We hope our comments help clarify the novelties of our work. If the reviewer believes their comments have been addressed, we would be very grateful if they could raise their score accordingly.

---

> > ### Author Response · Authors · 2022-08-02
> > **Author response (continued)**
> >
> > Specific critiques regarding strength of arguments:
> >
> > - **"Hard to map [spurious or core ftrs] to geometric concepts"** We agree, though we note that (i) there is a precedent for our simplified model [Khani et al. ] (ii) assuming disentangled features allows for quantifying the exact amount of spurious feature reliance via $NFS$ (iii) in practice, certain semantic features may indeed be disentangled either directly in the pixel space (i.e. background vs foreground) or even implicitly as information propagates to deeper layers in the neural network; in Salient ImageNet, deep neural nodes were identified as corresponding to core or spurious features for specific classes.
> >
> > - **"Strength of spurious correlations claims are unfounded/not quantified"** We note that the hyperparameter $\eta$ in section 3 directly corresponds to the degree of correlation between core and spurious features, which we see in the simplified setting having significant effects on the degree to which spurious features are used (Figures 2 and 3). We concur that proxying this quantity can be very challenging in practice, though we maintain that the insights from our theoretical analysis can still have utility, as they draw attention to how the specific spurious correlations at play really matter in terms of how and when models rely on them. Perhaps this analysis may inspire the community to make more efforts at identifying and measuring spurious correlations present in their data.
> >
> > - **"ImageNet-C and $\ell_\infty$ not explained"** Our theory focuses on distribution shifts in which only spurious features are disrupted, while core features remain intact, so to show that reduced natural distributional robustness of adversarially trained models occurs only in cases of distribution shifts that break spurious correlations. This is precisely the crucial difference between the distribution shifts introduced by ImageNet-C and ObjectNet, as ObjectNet primarily changes spurious features, while ImageNet-C corrupts the entire image, without disrupting correlations between spurious and core features. Therefore, our theory predicts reduced performance by adversarially trained models only for ObjectNet and not ImageNet-C, as our experiment shows.  Similarly, for $\ell_\infty$ AT models, we can see a clear distinction in our theoretical results from $\ell_1$ and $\ell_2$ AT, in that $\ell_\infty$ AT does not always lead to increased spurious feature reliance in our theory (Figures 2 and 3), which is then corroborated in the experiments.
> >
> > [1] Khani, Fereshte, and Percy Liang. "Removing spurious features can hurt accuracy and affect groups disproportionately." Proceedings of the 2021 ACM Conference on Fairness, Accountability, and Transparency. 2021.

---

> > > ### Comment · Reviewer_R1sj · 2022-08-06
> > > **Thank you for the detailed answer and clarifications**
> > >
> > > Thank you very much for the very detailed answer and clarifications. I believe your comments have made a strong case defending the novelty of your work as a more thorough and dedicated benchmarking of the initial observations performed in [38, 55]. However, as mentioned by [Reviewer gNwp](https://openreview.net/forum?id=MeYI0QcOIRg&noteId=4Hxn181EOL6) I still am not convinced with your explanations regarding Sec. 4.4. as I do not agree that the robustness community blindly uses "any" source of spurious correlation as a "scapegoat" to example adversarial vulnerability. In this regard, I would still encourage you to tone down a bit the claims in Sec 4.4. presenting your results as "some empirical evidence" that "not every form of spurious correlation" is responsible for $\ell_p$ adversarial weaknesses.
> > >
> > > Moreover, I still believe the writing of the paper could be greatly improved. Personally, I believe a thorough polishing of the text, mathematical derivations, and figures along would make for a much more appealing contribution.
> > >
> > > In any case, based on the rebuttal, and since I believe the merits of this work slightly outweigh its flaws, I have decided to increase my score to 5: Borderline accept.

---

> > > > ### Author Response · Authors · 2022-08-08
> > > > **Thank you, Extended polishing/clarity improvements to come if accpeted**
> > > >
> > > > We sincerely thank the reviewer for taking the time to read our (long) rebuttal, and increasing their score. We have made additional revisions to tone down the language in the recommended section, writing that our “experiment shows that, while some spurious correlations may cause adversarial vulnerability, others do the opposite: the picture is more nuanced than previously assumed”.
> > > >
> > > > We have also made efforts to improve the clarity of the manuscript, though space constraints make this difficult at the moment. If our paper is accepted, we plan on using the extra page to make more considerable improvements, such as greater explanation of our theoretical results, a conclusion, and more extended discussion of how our results relate and contrast with existing literature (i.e. clarifying that our result emphasizes how important the specific type of spurious feature is with respect to how relying on that feature effects multiple notions of robustness). With the extra space, we hope to encourage future work that investigates exactly how and when certain spurious features can be used effectively as context to improve model performance and robustness.
> > > >
> > > > Once again, thank you for the score increase. If you concur that our result is novel and significant enough to inspire closer attention at the intriguing and potentially surprising ways that different notions of robustness interact, we would extremely appreciate an additional score increase. Especially thanks to the detailed feedback and discussion with you and Reviewer gNwp, we believe that will be able to deliver a clear and impactful message in the full manuscript (if accepted).

---

### Meta-Review · Area_Chair_e4RL · 2022-08-26

**Recommendation:** Accept
**Confidence:** Certain

**Metareview:**

All reviewers recommend accepting the paper, so I will follow their suggestion - congratulations!

However, I should add that I do not find the arguments in the paper fully convincing. Earlier work has shown that adversarially trained models behave like "standard" ImageNet models (i.e., models trained without a robustness-enhancing method) on ObjectNet, which is one of the datasets in this paper - see https://arxiv.org/abs/2007.00644 . In these results, there is no trade-off between adversarial robustness and robustness on ObjectNet (on the other hand, adversarial robustness also doesn't help on ObjectNet). I encourage the authors to engage with the prior work on out-of-distribution generalization more deeply because this will help the reader understand how the new results here relate to other findings in out-of-distribution generalization.

**Award:**

No

---

### Decision · Program_Chairs · 2022-09-14

Accept